# CAUSAL REINFORCEMENT LEARNING FOR SPATIO-TEMPORAL POINT PROCESSES

## ABSTRACT

Spatio-temporal event sequences are increasingly accessible in various domains such as earthquake forecasting, crime prediction, and healthcare management. These data sources present unique challenges, as they involve both spatial and temporal dimensions, with event sequences exhibiting intricate dependencies over time and space. Neural network-based spatio-temporal point processes offer a sophisticated framework for modeling such event data. Conventional maximum likelihood estimation (MLE) of such data may lead to inaccurate predictions due to model-misspecification and compounding prediction errors. On the other hand, reinforcement learning frameworks, which treat event generation as actions and learn a policy to mimic event generation, may alleviate the training/test discrepancy issue. Current reinforcement learning of point processes may have prohibitively poor exploration efficiency. In this paper, we propose the Causal learning improved Reinforcement Learning Spatio-Temporal Point Process (CRLSTPP) framework, which can mitigate the issue of compounding prediction errors and improve exploration efficiency at the same time. Experiments on both synthetic data and real-world data validate the superiority of the proposed model.

## 1 INTRODUCTION

Asynchronous spatio-temporal event data, such as crime records, traffic flow patterns, and earthquake occurrences, are widely utilized across various domains including predictive policing Zhu et al. (2021b); Zhou et al. (2022); Mohler et al. (2011), urban transportation Yuan et al. (2023); Zhu et al. (2021a), and seismology Chen et al. (2020); Ogata (1988). Different from regularly sampled spatial or temporal data, the asynchronous sequence contains events with spatial coordinates and time stamps that occur in continuous space and time. Those event sequences often demonstrate notable self-exciting patterns. For instance, a crime event often leads to further incidents in neighboring areas and time frames. It is critical to model these complex underlying dynamics such that accurate predictions can be made sequentially given the history Baddeley et al. (2007); Chen et al. (2020); Diggle (2006).

Spatio-temporal point processes (STPPs) are well suited for modeling event data of this nature. Typically, an STPP model is characterized by an intensity function to indicate the instantaneous rate of an event occurring at a specific location and time given all historical events. Classic Spatio-temporal point process models Moller & Waagepetersen (2003); Ogata (1988; 1998) leverage prior process knowledge to make parametric or semi-parametric assumptions on the intensity function. Although these models tend to perform well when the underlying parametric assumptions align with the actual processes, in scenarios where the genuine generative process of events remains obscure, these parametric assumptions can become excessively restrictive and do not represent the underlying dynamics well. Alternatively, variations of point processes have been proposed to model intensity functions by incorporating nonparametric forms Yuan et al. (2023), automatic integrationZhou & Yu (2024), neural ordinary differential equations Chen et al. (2020), and recurrent neural networks Yang et al. (2018).

Conventionally, STPP models are trained via supervised learning, and model parameters are estimated by maximizing the log-likelihood (MLE) of observed event sequences. However, models learned via MLE at training time may be misspecified and may encounter compounding prediction errors over long horizons of composed forecasting Arora et al. (2022); Tan et al. (2018); Li et al. (2023), where

prediction errors made during testing accumulate in multiple successive forecasting steps and lead to states distant from training data. Alternative methods address the training/test discrepancy problem by employing reinforcement learning (RL) techniques Ranzato et al. (2015); Ding & Soricut (2017); Bahdanau et al. (2016); Li et al. (2018). However, RL-based approaches may encounter issues with the exploitation efficiency during the action sampling stage.

In this paper, we propose a novel causal reinforcement learning framework for modeling the spatio-temporal point process. Firstly, we introduce a new approach by examining spatio-temporal point processes from a reinforcement learning perspective. More specifically, the action is the next event's location and time, the state consists of the parameters that characterize the underlying spatial temporal distribution, and the generation of each event can be interpreted as an action taken by a stochastic policy. The observed sequences will be seen as the expert policy, and our goal is to use an MMD-based reward to minimize the discrepancy between the learner policy and the expert policy. Such a framework alleviates the training/testing discrepancy and reduces compounding prediction errors.

Secondly, to further enhance exploration efficiency, we measure the causal influence of actions on the state and choose actions that are predicted to have a higher causal impact. Similar to the self-exciting point process, the arrival action (event) will influence the underlying distribution. Our goal is to measure this causal impact and to choose the action with the higher casual influence score. Additionally, unlike traditional discrete actions, our actions consist of event time stamps and spatial locations, which are continuous. To tackle this issue, our method involves grid sampling to cover the action space comprehensively; after that, we select the action that has the maximum deviation from the expected outcome, enabling the policy to take the action that significantly improves the exploration efficiency. Comprehensive experiments on both synthetic and real-world data demonstrate that our proposed algorithm outperforms the state-of-the-art.

## 2 RELATED WORK

**Spatial and Temporal Point Process** Existing approaches in spatio-temporal point process modeling such as kernel function-based models Zhang et al. (2023); Dong et al. (2023b); Zhu et al. (2021c); Ilhan & Kozat (2020); Dong et al. (2023a) often rely on simplified parametric assumptions to represent the underlying dynamics of events such as seismic activities. Most of these solutions are based on the intensity paradigm, which might not be sufficiently flexible for different applications. Recently, some generative models Li et al. (2021b); Yuan et al. (2023); Xiao et al. (2017) have been proposed to solve the training/testing discrepancy issue. However, the training complexity of generative models is significant, which requires a substantial number of iterations and meticulous parameter tuning to ensure that both neural networks achieve a state of equilibrium. Recent efforts aim to uncover Granger causality in temporal point processes Jalaldoust et al. (2022); Xu et al. (2016).

A reinforcement framework Li et al. (2018) is proposed to model a temporal point process for sequences with only temporal information. An imitation learning-based algorithm Zhu et al. (2021b) has been proposed to train a spatio-temporal point process Zhu et al. (2021b). The work relies on fully connected neural networks to parameterize a mixture of heterogeneous Gaussian diffusion kernels for modeling spatio-temporal dependencies. These kernels capture complex spatial patterns, but the model does not focus on capturing long-term temporal dependencies. Moreover, models based on imitation learning also have some limitations due to their constrained exploratory capabilities Ali (2021). In our work, we incorporate the recurrent mixture density network (RMDN) Bazzani et al. (2016) structure, which effectively captures long-term dependencies between events.

**Causality Improved Reinforcement Learning** Incorporating causality information to enhance the performance of reinforcement learning (RL) frameworks is an emerging trend Zeng et al. (2023). Recent studies have augmented RL performance by integrating causal learning in various ways. Examples include utilizing the causal information to enhance sample efficiency in the online setting Wang et al. (2021), using causality action inference as a part of reward bonus Seitzer et al. (2021), and exploring actions with causal influence Lu et al. (2022); Seitzer et al. (2021); Bica et al. (2021). Additionally, to improve the interpretability of the reinforcement learning model, some existing works focus on learning the causal graph from the reinforcement learning process Ding et al. (2022); Li et al. (2021a). In this paper, we introduce causal information-based action exploration RL to the domain of spatio-temporal point processes (STPP) for the first time, and we focus on causal information detection to improve event prediction.

## 3 PRELIMINARIES

### 3.1 SPATIO TEMPORAL POINT PROCESS

A spatio-temporal point process is a stochastic process consisting of a sequence of events. The $i^{th}$ event can be represented as $a_i = (\mathbf{z}_i, t_i)$ where $t_i \in \mathbb{R}^+$ is the event occurrence time and $\mathbf{z}_i \in \mathbb{R}^d$ is the event location in a $d$-dimensional space. Let $\mathcal{H}_t$ denote the history up to time $t$, $\mathcal{H}_t = \{(\mathbf{z}_1, t_2), (\mathbf{z}_2, t_2), \ldots, (\mathbf{z}_n, t_n) | t_n < t\}$. The spatio-temporal point process can be characterized by the conditional intensity function:

$$\lambda(\mathbf{z}, t | \mathcal{H}_t) = \lim_{\Delta t, \Delta \mathbf{z} \downarrow 0} \frac{\mathbb{P}(t_i \in [t, t + \Delta t], \mathbf{z}_i \in B(\mathbf{z}, \Delta \mathbf{z}) | \mathcal{H}_t)}{|B(\mathbf{z}, \Delta \mathbf{z})| \Delta t}, \tag{1}$$

where the $B(\mathbf{z}, \Delta \mathbf{z})$ denotes a Euclidean ball centered at $\mathbf{z} \in \mathbb{R}^d$ with radius $\Delta \mathbf{z}$, and the $|\cdot|$ is the Lebesgue measure.

The likelihood of observing an event at time $t$ and location $\mathbf{z}$ is defined by:

$$\log p(\mathcal{H}) = \sum_{i=1}^{n} \log \lambda(\mathbf{z}_i, t_i | \mathcal{H}_{t_i}) - \int_0^T \int_{\mathbb{R}^d} \lambda(\mathbf{z}, t | \mathcal{H}_t) d\mathbf{z} dt. \tag{2}$$

The spatio-temporal point processes introduce additional complexity to the temporal ones since they require modeling the correlation between events across space. Training general STPPs with maximum likelihood poses a challenge due to the need to solve a complex multivariate integral. For computational tractability, approximations or numerical methods are often used to evaluate the integrals, especially over continuous space.

### 3.2 POLICY GRADIENT REINFORCEMENT LEARNING

Policy gradient is a category of reinforcement learning techniques where parameterized policies are learned to maximize the cumulative long-term reward through gradient descent. In contrast to value-based approaches, policy gradient methods focus on the direct optimization of the policy that guides the agent's actions. The objective in policy gradient methods is to maximize the expected return, defined as:

$$J(\theta) = \mathbb{E}_{\tau \sim \pi_\theta}[R(\tau)], \tag{3}$$

where $\theta$ represents the parameters of the policy $\pi$, $\tau$ denotes the trajectory, $R(\tau)$ is the total reward for trajectory $\tau$. Additionally, the gradient of $J(\theta)$ with respect to $\theta$ is given by the policy gradient theorem:

$$\nabla_\theta J(\theta) = \mathbb{E}_{\tau \sim \pi_\theta} \left[ \sum_{\tau=0}^{T} \nabla_\theta \log \pi_\theta(a_\tau | s_\tau) R(\tau) \right], \tag{4}$$

where $a_t$ and $s_t$ are the action and state at time step $t$, respectively, and $T$ is the time horizon of the trajectory.

## 4 PROPOSED MODEL

### 4.1 REINFORCEMENT LEARNING OF STPP

In this paper, we focus on modeling intricate distributions of continuous time and space. We employ the causal information-based reinforcement learning (RL) framework to model the STPP by integrating the ideas from policy gradient reinforcement learning and casual influence detection Seitzer et al. (2021).

We regard the observed spatio-temporal patterns as a series of actions enacted by the expert policy $\pi_E$. Let $\xi = \{(\mathbf{z}_1, t_1), (\mathbf{z}_2, t_2), \ldots, (\mathbf{z}_m, t_m)\}$ denote a spatio-temporal trajectory of $m$ events from the expert, where $(\mathbf{z}_i, t_i)$ indicates an event at location $\mathbf{z}_i$ and at time $t_i$. The event count $m$ may differ among various trajectories (sequences). Each event in the trajectory $\xi$ can be seen as an action sampled from the expert's spatio-temporal strategy $\pi_E$. Given a set of expert trajectories $E = \{\xi_1, \xi_2, \ldots, \xi_k, \ldots | \xi \sim \pi_E\}$, fitting a spatio-temporal point process to $E$ can be seen as finding a

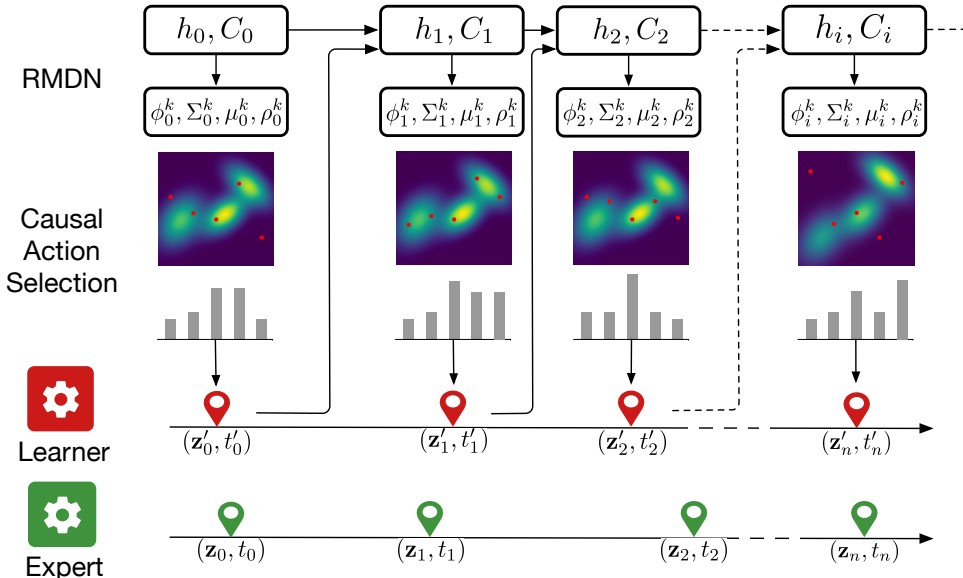

Figure 1: Our reinforcement learning framework is enhanced by causal learning. For each recurrent unit (excluding the initial one), the input consists of the output from the previous unit and the action generated by that previous unit. The observed sequences are treated as the expert policy, and the reward is used to minimize the discrepancy between the learner's policy and the expert policy.

learner policy $\pi_L$ that is capable of generating a new set of trajectories $\widetilde{E} = \{\widetilde{\xi}_1, \widetilde{\xi}_2, \ldots, \widetilde{\xi}_k, \ldots | \widetilde{\xi}_k \sim \pi_L\}$ close to $E$.

Formally, consider a series of past events $\xi_t = \{(\mathbf{z}_1, t_1), \ldots, (\mathbf{z}_i, t_i)\}_{t_i < t}$. Within this framework, the stochastic learner policy $\pi_L$ samples a location and a time interval and generates the next event $(\mathbf{z}_{i+1}, t_{i+1})$. Following this sampling, the trajectory $\xi_t$ updates to $\xi_{t+1} = \{(\mathbf{z}_1, t_1), \ldots, (\mathbf{z}_i, t_i), (\mathbf{z}_{i+1}, t_{i+1})\}$ and the reward $r(\mathbf{z}_{i+1}, t_{i+1})$ will be calculated. Provided the reward function $r(\mathbf{z}, t)$, the optimal policy $\pi_L^*$ can be deduced directly through:

$$\pi_L^* = \arg \max_{\pi_L \in P} J(\pi_L) := \mathbb{E}_{\iota \sim \pi_L} \left[ \sum_{i=1}^{N^\iota} r(\mathbf{z}_i, t_i) \right], \tag{5}$$

where $P$ is the set of all candidate spatio-temporal policies $\pi_L$, $\{(\mathbf{z}_1, t_1), \ldots, (\mathbf{z}_{N^\iota}, t_{N^\iota})\}$ is a specific event sampled from policy $\pi_L$, and $N^\iota$ may differ for various samples.

As shown in Figure 1, we adopt a recurrent mixture density network (RMDN)Bazzani et al. (2016) enhanced by stochastic neuronsHochreiter & Schmidhuber (1997) as the policy network to generate spatio-temporal events. To improve the efficiency of exploration, we adopt conditional mutual information (CMI) to measure the influence of potential actions and prioritize actions that are predicted to have a higher causal impact on the state.

## 4.2 Policy Network

The policy $\pi_L$ needs to be sufficiently adaptable and comprehensive to encapsulate the intricate patterns of spatio-temporal point processes as observed in the expert's data. Thus, we utilize RMDN enhanced by stochastic neurons to accommodate the non-linear and extensive sequential dependencies inherent in the data.

Specifically, we assume the next time interval of an event follows Rayleigh distribution and the location of an event follows the mixture of Gaussian distribution. The parameters of those distributions depend on the hidden state $h_i$ at time $t$. Each recurrent unit corresponds to an event in the sequence,

and the unit is represented by the following functions:

$$f_i = \varphi(W_f \cdot [h_{i-1}, x_i] + b_f), \qquad p_i = \varphi(W_p \cdot [h_{i-1}, x_i] + b_p), \qquad (6)$$

$$o_i = \varphi(W_o \cdot [h_{i-1}, x_i] + b_o), \qquad \widetilde{C}_i = \tanh(W_C \cdot [h_{i-1}, x_i] + b_C), \qquad (7)$$

$$C_i = f_i * C_{i-1} + p_i * \widetilde{C}_i, \qquad h_i = o_i * \tanh(C_i), \qquad (8)$$

where $\varphi$ is the logistic sigmoid function and the $f_t, p_t, o_t, C_t, h_t$ represent the forget gate, input gate, output gate, memory cell, and hidden state, respectively. And the $x_i = \{\mathbf{z}_i, t_i\}$ will be the action generated by the previous unit.

To generate nonnegative-valued random time intervals, we adopt the Rayleigh distribution, and the probability density function can be written as:

$$PDF(\psi(h_{i-1}), \alpha) = \frac{\psi(h_{i-1})}{\alpha^2} e^{-\frac{\psi(h_{i-1})^2}{2\alpha^2}}, \qquad (9)$$

where $\psi$ is a nonlinear mapping from $\mathbb{R}^d$ to the parameter space of the learner probability distribution, and the $\alpha$ is the scale parameter of the Rayleigh distribution. With that, we can generate the time intervals as:

$$t_i \sim \pi(t|\psi(h_{i-1})). \qquad (10)$$

After that, the mixture density network will take the hidden state $h_i$ as the input and generate the parameters of the Gaussian mixture model, from which we sample a location $\mathbf{z}_i$. We can reparameterize the model as $\{(\mu_i^k, \phi_i^k, \sigma_i^k, \rho_i^k)\}_{k=1}^K$, where $\mu_i^k, \phi_i^k, \sigma_i^k$, and $\rho_i^k$ are the 2D mean position, the weight, the 2D variance and the correlation of the k-th Gaussian component, respectively. The mixture model is therefore defined as follows:

$$y_i = \{(\tilde{\mu}_i^k, \tilde{\phi}_i^k, \tilde{\sigma}_i^k, \tilde{\rho}_i^k)\}_{k=1}^K = W_y \cdot h_i + b_y, \qquad (11)$$

where $w_y$ and $b_y$ are the weights to be estimated. We assume that the mixture Gaussian distribution will decay with time, and the 2D location of the STPP can be sampled as:

$$\mathbf{z}_i \sim \pi(\mathbf{z}|y_i \cdot e^{-\beta(t_i - t_{i-1})}). \qquad (12)$$

The stochastic action-generating component within our RMDN is designed to emulate the inherent stochastic mechanisms of spatio-temporal point processes. The parameters of the spatial temporal distributions such as $\mu, \sigma$ and $\rho$ are the state in our reinforcement learning framework. Furthermore, once an action is generated, it will influence the underlying intensity distribution and have a cascading effect on future events, similar to the dynamics observed in a self-exciting point process.

### 4.3 Causal Information Improved Action Selection

In this paper, we introduce the causal information based action selection method to enhance the exploration efficiency of the RL framework. Due to the self-exciting properties of spatio-temporal point processes (STPP), in our RL framework, once an action is selected, it will influence the subsequent state. Different actions exhibit varying degrees of influence on the state. To improve the efficiency of exploration, our approach uses conditional mutual information to measure the influence and prioritize actions expected to exert the most significant causal influence on the next state.

Firstly, unlike traditional discrete actions, our actions consist of event times and spatial locations, which are continuous. We employ grid sampling to ensure comprehensive sampling while preserving the distribution's characteristics. We discretize the region of interest into $n \times n$ grid cells. Then we generate actions from the learner policy $\pi_L$ iteratively until $q$ cells are covered by at least one event sample. Finally, we extract $q$ samples and each of them is located in a distinct grid cell.

Once an action ($a_i = (\mathbf{z}_i, t_i)$) is generated, the hidden states $h$ and the parameters of RMDN will change. Thus, the underlying spatio-temporal intensity will change. To encourage the action exploration efficiency, we choose the possible event (action) that results in maximal conditional mutual information deviation from the expectation (see Appendix A for details). The process of choosing the action can be described as:

$$a^* = \underset{a \in \{a^1, \dots, a^z\}}{\arg\max} \left( D_{KL} \left( p(y_i|y_{i-1}, a) \,\middle\|\, \frac{1}{q} \sum_{k=1}^q p(y_i|y_{i-1}, a^q) \right) \right), \qquad (13)$$

where the $a \in \{a^1, \ldots, a^q\} \sim \pi_L$, and $y$ is the output of the RMDN as described in 11. Intuitively, the selected action will be the one that causes the maximum deviation from the expected next state among all possible actions.

## 4.4 REWARD FUNCTION

For spatio-temporal point processes, the complex patterns within the sequence cannot be adequately scored using the L1 distance reward. Therefore, to effectively compare the STPP sequence, we adopt an MMD reward Kim & Pineau (2013) function with a Gaussian kernel. The MMD measures the distance between two sequences, by mapping data points into a reproducing kernel Hilbert space (RKHS) using a kernel function and computing the difference between their mean embeddings. The definition is the following equation:

$$\text{MMD}^2(\xi_L, \xi_E) = \frac{1}{M_L^2} \sum_{i=1}^{M_L} \sum_{j=1}^{M_L} k(a_i, a_j) + \frac{1}{M_E^2} \sum_{i=1}^{M_E} \sum_{j=1}^{M_E} k(e_i, e_j) - \frac{2}{M_L M_E} \sum_{i=1}^{M_L} \sum_{j=1}^{M_E} k(a_i, e_j),$$
(14)

where $a_i = (\mathbf{z}_i, t_i)$ is the learner policy action, $e_i$ is the expert action, $M_L$ is the event count of learner, and $M_E$ is the event count of expert. Thus, our reward can be written as:

$$r(a) \propto -\sum_{u=1}^{N_E} \sum_{v=1}^{N_L} \text{MMD}^2(\xi_L^v, \xi_E^u),$$
(15)

where $N_E$, $N_L$ are the number of sequences generated by the expert and the learner, respectively.

## 4.5 POLICY GRADIENT

When utilizing Policy Gradient methods for training, the focus often lies on minimizing the divergence $D(\pi_E, \pi_L, \mathcal{H})$. However, it is equally viable to minimize $D(\pi_E, \pi_L, \mathcal{H})^2$ since the squaring function is a monotonic transformation. This approach falls within the reinforcement learning paradigm described in formulation 4. Employing the policy gradient technique, the gradient $\nabla_L D(\pi_E, \pi_L, \mathcal{H})^2$ is computed as:

$$\nabla_L J(L) \approx \frac{1}{M_E} \sum_{j=1}^{M_E} \left[ \sum_{i=1}^{n_j} \left( \nabla_L \log \pi_L(a_i) \cdot r(a_i) \right) \right],$$
(16)

where $\nabla_L \log \pi_L(a_i)$ denotes the gradient of the log-likelihood of an action $a_i$, drawn from a rollout sample $\xi = \{(\mathbf{z}_1, t_1), (\mathbf{z}_2, t_2), \ldots, (\mathbf{z}_{N^\iota}, t_{N^\iota})\}$ under the learner's policy $\pi_L$. If we assume that spatial process and temporal process are independent of each other, we can simply multiply the two distributions to obtain the log-likelihood of action $a_i$.

Also, to reduce the variance of the gradient, we adopt the Mini-batch Reinforcement Learning Li et al. (2018). For each training iteration, firstly, we sample a minibatch of $M_l$ trajectories of events from the expert policy $\pi_E$ and a mini-batch of $M_e$ trajectories from the learner policy $\pi_L$. These trajectories are used to compute the policy gradient $\nabla_\theta D(\pi_E, \pi_L, \mathcal{H})^2$, which is an average over the mini-batch gradients weighted by the estimated optimal policy $\pi_L^*$. The log-likelihood is obtained by multiplying the spatial distribution and temporal distribution. Additionally, the policy parameters are updated by moving in the gradient direction scaled by a learning rate $\eta_l$. This approach can diminish the divergence between learner and expert policy distributions, thereby training the model to be close to the trajectories in STPP.

## 5 EXPERIMENT

In this section, we conduct experiments to address the following questions:

**Q1:** How does the proposed model perform compared to existing baseline methods?
**Q2:** What is the effect of causal learning on improving exploration efficiency?
**Q3:** Has the model addressed the compounding prediction errors issue?
**Q4:** How to attain a more profound comprehension of the RL-based STPP framework?

Table 1: Evaluation of performance in predicting both the time and location of the next 1 event.

| Model | Synthetic Temporal | Synthetic Spatial | Earthquake Temporal | Earthquake Spatial | 911 Call Temporal | 911 Call Spatial | Crime Temporal | Crime Spatial |
|---|---|---|---|---|---|---|---|---|
| CNF | - | $0.102_{\pm 0.00}$ | - | $7.212_{\pm 0.08}$ | - | $0.218_{\pm 0.01}$ | - | $0.191_{\pm 0.00}$ |
| THP | $0.161_{\pm 0.01}$ | - | $0.412_{\pm 0.03}$ | - | $0.230_{\pm 0.04}$ | - | $0.601_{\pm 0.04}$ | - |
| NEST | $0.162_{\pm 0.01}$ | $0.099_{\pm 0.01}$ | $0.415_{\pm 0.03}$ | $7.011_{\pm 0.61}$ | $0.287_{\pm 0.02}$ | $0.194_{\pm 0.01}$ | $0.672_{\pm 0.03}$ | $0.193_{\pm 0.00}$ |
| NSTPP | $0.179_{\pm 0.00}$ | $0.097_{\pm 0.00}$ | $0.492_{\pm 0.01}$ | $6.983_{\pm 0.11}$ | $0.232_{\pm 0.01}$ | $\mathbf{0.173}_{\pm 0.00}$ | $0.611_{\pm 0.02}$ | $\mathbf{0.099}_{\pm 0.00}$ |
| DeepSTPP | $0.198_{\pm 0.01}$ | $0.106_{\pm 0.00}$ | $0.427_{\pm 0.02}$ | $\underline{6.747}_{\pm 0.53}$ | $0.202_{\pm 0.01}$ | $0.188_{\pm 0.00}$ | $\underline{0.578}_{\pm 0.01}$ | $0.118_{\pm 0.00}$ |
| DSTPP | $0.171_{\pm 0.02}$ | $\mathbf{0.084}_{\pm 0.01}$ | $\mathbf{0.390}_{\pm 0.02}$ | $6.915_{\pm 0.75}$ | $0.217_{\pm 0.02}$ | $\underline{0.178}_{\pm 0.02}$ | $\mathbf{0.564}_{\pm 0.04}$ | $\underline{0.102}_{\pm 0.02}$ |
| CRLSTPP | $\mathbf{0.152}_{\pm 0.02}$ | $\underline{0.089}_{\pm 0.00}$ | $\underline{0.393}_{\pm 0.02}$ | $\mathbf{6.608}_{\pm 0.58}$ | $\mathbf{0.197}_{\pm 0.01}$ | $0.182_{\pm 0.02}$ | $0.581_{\pm 0.03}$ | $0.127_{\pm 0.01}$ |

Table 2: Evaluation of performance in predicting both the time and location of the next 20 events.

| Model | Synthetic Temporal | Synthetic Spatial | Earthquake Temporal | Earthquake Spatial | 911 Call Temporal | 911 Call Spatial | Crime Temporal | Crime Spatial |
|---|---|---|---|---|---|---|---|---|
| CNF | - | $3.25_{\pm 0.37}$ | - | $302.53_{\pm 29.31}$ | - | $6.03_{\pm 0.77}$ | - | $4.41_{\pm 0.26}$ |
| THP | $4.35_{\pm 0.61}$ | - | $11.13_{\pm 2.06}$ | - | $8.014_{\pm 1.83}$ | - | $15.92_{\pm 3.10}$ | - |
| NEST | $\underline{4.13}_{\pm 0.47}$ | $\underline{2.55}_{\pm 0.35}$ | $\underline{9.44}_{\pm 1.22}$ | $231.25_{\pm 28.34}$ | $\underline{7.42}_{\pm 1.10}$ | $4.70_{\pm 0.48}$ | $15.59_{\pm 2.36}$ | $3.98_{\pm 0.97}$ |
| NSTPP | $4.60_{\pm 0.31}$ | $3.71_{\pm 0.30}$ | $11.17_{\pm 1.24}$ | $274.14_{\pm 25.53}$ | $8.49_{\pm 1.02}$ | $5.51_{\pm 0.81}$ | $17.24_{\pm 2.14}$ | $4.71_{\pm 0.61}$ |
| DeepSTPP | $4.22_{\pm 0.51}$ | $3.51_{\pm 0.42}$ | $11.23_{\pm 1.01}$ | $268.09_{\pm 31.04}$ | $8.51_{\pm 1.07}$ | $5.72_{\pm 0.92}$ | $16.71_{\pm 2.46}$ | $4.55_{\pm 0.91}$ |
| DSTPP | $4.28_{\pm 0.72}$ | $3.42_{\pm 0.60}$ | $10.59_{\pm 1.46}$ | $\underline{228.12}_{\pm 33.47}$ | $8.77_{\pm 1.38}$ | $5.38_{\pm 1.19}$ | $\underline{15.98}_{\pm 3.06}$ | $\underline{3.79}_{\pm 0.89}$ |
| CRLSTPP | $\mathbf{3.29}_{\pm 0.84}$ | $\mathbf{2.02}_{\pm 0.34}$ | $\mathbf{8.02}_{\pm 1.02}$ | $\mathbf{201.28}_{\pm 22.19}$ | $\mathbf{7.02}_{\pm 1.14}$ | $\mathbf{3.71}_{\pm 0.73}$ | $\mathbf{13.04}_{\pm 1.82}$ | $\mathbf{3.28}_{\pm 0.83}$ |

## 5.1 DATASET

We utilize one synthetic data and three real-world datasets exhibiting diverse temporal dynamics across various domains, and further details can be found in Appendix B. Our code will be made available on GitHub upon the acceptance of the paper.

**Synthetic Data:** Sequences were generated by a self-exciting process with event locations sampled from a mixture of Gaussian distributions which $K = 3$. **Boston Crime:** Motor vehicle accident event happened in Boston during 2015.06.14 - 2018.09.03[1] **911 Call:** The fire 911 calls reported in Montgomery County, PA, spanning from December 2015 to December 2018, with complete location data.[2] **Earthquake:** Earthquakes in the West U.S. with a magnitude of a least 2.5 from 1990 to 2020 were recorded by the U.S. Geological Survey.[3]

## 5.2 BASELINE

We evaluate our proposed model by comparing it to several spatial models, temporal models, and spatio-temporal point process models across various datasets. **Neural Embedding Spatio-Temporal (NEST) Zhu et al. (2021b):** Mixture Gaussian diffusion kernels based spatio-temporal process model, learning parameters by imitation learning. **Spatio-Temporal Diffusion Point Processes(DSTPP) Yuan et al. (2023):** Diffusion model-based framework for spatio-temporal point processes, enabling the learning of complex joint distributions. **Neural Spatio-temporal Point Process (NSTPP) Chen et al. (2020):** Neural ODE-based STPP model, utilizing Neural Jump SDEs for parameterizing temporal intensity and continuous-time normalizing flows for spatial PDFs. **Neural Point Process for Learning Spatiotemporal Event Dynamics (DeepSTPP) Zhou et al. (2022):** Variational inference based deep STPP model. **Continuous normalizing flow (CNF) Chen et al. (2018):** Neural ordinary differential equations based spatial prediction model. **Transformer Hawkes Process (THP) Zuo et al. (2020):** Transformer based temporal point process model.

---

[1]https://www.kaggle.com/datasets/AnalyzeBoston/crimes-in-boston

[2]https://www.kaggle.com/code/mohammadhy/analysis-911-call

[3]https://www.usgs.gov/programs/earthquake-hazards

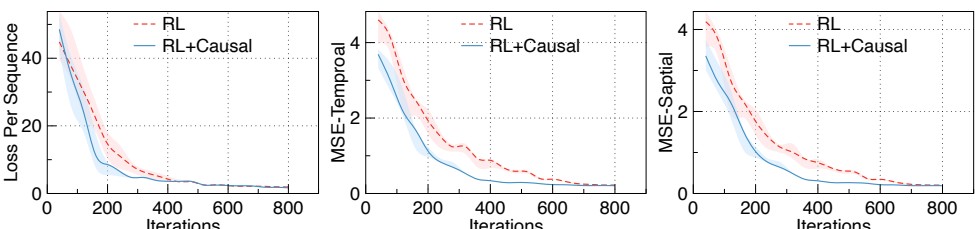

Figure 2: The comparison of exploration efficiency with/without causal action selection.

### 5.3 EVALUATION METRICS

We assess the models' performance from two angles: (1) Computing mean square error (MSE) for the one-step ahead prediction of temporal and spatial (2) To demonstrate its effectiveness in addressing compounding prediction errors, we calculate the accumulative MSE for 20 steps ahead prediction. We refer readers to Appendix D for more details on the evaluation metrics, parameters setting, and hyperparameter selection experiments.

### 5.4 PERFORMANCE

Table 1 and 2 present the comprehensive performance of models in terms of prediction accuracy and ability to address the compounding prediction error issues, respectively. (See Appendix C for additional performance results on methods that focus exclusively on spatial or temporal aspects.) Additionally, Figure 2 shows the comparison of reinforcement learning exploration efficiency with/without causal learning. Based on these results, we can draw the following conclusions:

**Our model significantly alleviates compounding prediction errors.** The 20-steps ahead prediction MSE results demonstrate our model has superior performance in mitigating the compounding prediction errors. This success is attributed to the reinforcement learning strategy, which can learn robust policies that account for dependencies between sequential decisions. Furthermore, instead of making point predictions at each time step independently, RL learns a policy that maps states to actions, which reduces the errors arising from historical predictions. This approach stands in contrast to MLE and other methods, which do not inherently consider the sequential decision-making process and the potential accumulation of prediction errors. Additionally, our model achieves good performance in one-step prediction, demonstrating that the introduction of reinforcement learning and the focus on multi-step forecasting do not compromise the accuracy of short-term predictions.

**Following actions with the most Causal influence can improve the exploration efficiency.** From the convergence experiments depicted in Figure 2 , it is obvious that the model exhibits higher exploration efficiency when employing a causal influence-based action selection method. This enhanced efficiency can be attributed to the causal influence-based approach enables the model to prioritize actions that are more likely to influence the outcome of interest directly. By understanding and leveraging the causal relationships within the environment, the model can avoid wasting effort on exploring actions with minimal or no impact on the desired outcome, leading to more efficient exploration.

**Appropriate parametric assumptions are essential for the effectiveness of point processes.** These assumptions about the parameter space can significantly influence the model's ability to accurately capture the underlying temporal or spatial structures inherent in the data. When the assumptions align well with the real-world phenomena being modeled, point processes can offer precise predictions, uncover hidden patterns, and provide deep insights into the dynamics of complex systems. Conversely, if the parametric assumptions are not well-suited to the data, the model may fail to converge, exhibit poor predictive performance, or even misinterpret the causal relationships within the data.

**Intensity recovery on synthetic data.** To better understand the mixture density network mechanism, we examine the location spatial distribution at different stages of the recovery iterations. The figures 3 illustrate the intensity recovery performance of our models in discrete-space scenarios, Initially, the spatial distribution contains 3 distributions with blurred appearance. This stage represents the early phase of training, where the separation between regions is still not clear. As the reinforcement

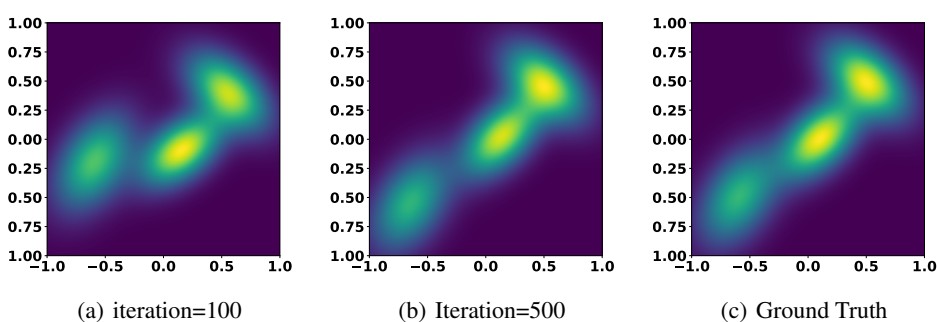

Figure 3: Intensity Recovery On the Synthetic Dataset.

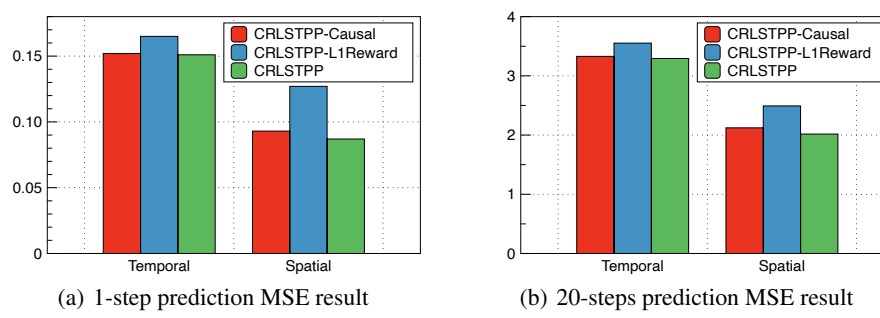

Figure 4: Ablation study on the action selection and reward function.

learning process continues, the spatial distribution becomes more concentrated and the noise level is significantly reduced. This suggests that the model is increasingly learning the underlying patterns. In the final stages, the spatial distribution is close to the ground truth, indicating that the model has effectively mastered the generative process of the spatial distribution.

## 5.5 ABLATION STUDY

In our ablation study, we evaluated the impact of two critical components: the causal inference-based action selection, and MMD-based reward. The baseline model CRLSTPP, which includes all components, achieved the highest performance metrics. When the MMD reward was modified to the L1 distance-based reward, we observed a significant drop in accuracy, indicating the essential role of the reward design in this RL model. Excluding causal inference-based action selection resulted in a moderate decline in accuracy, this is because the causal inference-based action selection method can improve the exploration efficiency significantly but not influence the accuracy too much.

## 6 CONCLUSION

We propose a novel causal reinforcement learning framework for modeling spatio-temporal event sequences. Under this framework, the policy is designed as a special recurrent mixture density network (RMDN), enabling the sequential sampling of discrete events. We update the policy by maximizing the reward which correlates with the disparity between the generated sequences and observed sequences. This approach effectively reduces the error accumulation in likelihood-based methodologies for multi-step forecasting. Furthermore, to improve the exploration efficiency of our reinforcement learning model, we have implemented a strategy that utilizes causal information to optimize event generation. This is achieved by selecting the event (action) that induces the maximum deviation in mutual information from its expected values. Experiments on synthetic and real-world datasets demonstrated our model's methodological robustness and superior performance in modeling complex spatio-temporal event data. This framework can be extended to accommodate more complex causal structures and explore the integration of additional data types in future works.

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

# A    CAUSAL INFLUENCE DETECTION BASED ON CMI

Conditional Mutual Information (CMI) $I(X; Y \mid Z)$ measures the amount of information that two random variables $X$ and $Y$ share, given the knowledge of a third variable $Z$. In the context of reinforcement learning, we can think of $X$ as the action $a$, $Y$ as the current state $s$, and $Z$ as the previous state $s_{i-1}$

CMI is given by the formula:

$$I(S_i; A \mid S_{i-1}) = H(S_i \mid S_{i-1}) - H(S_i \mid S_{i-1}, A)$$

where:

- $H(S_i \mid S_{i-1})$ is the conditional entropy of $S_i$ given $S_{i-1}$.

- $H(S_i \mid S_{i-1}, A)$ is the conditional entropy of $S_i$ given $S_{i-1}$ and $A$.

To understand the relationship between CMI and the KL divergence used in our action selection process, we delve into the derivation. The KL divergence $D_{KL}(P \| Q)$ between two distributions $P$ and $Q$ measures how one probability distribution diverges from a expected probability distribution. In our context, $P$ is $p(s_i \mid s_{i-1}, a)$ and $Q$ is the average distribution $\frac{1}{Z} \sum_{k=1}^{Z} p(s_i \mid s_{i-1}, a^z)$. The KL divergence is expressed as:

$$D_{KL}\left( p(s_i \mid s_{i-1}, a) \middle\| \frac{1}{Z} \sum_{k=1}^{Z} p(s_i \mid s_{i-1}, a^z) \right)$$

This KL divergence measures how much information the action $a$ provides about the state $s_i$, conditioned on $s_{i-1}$.

CMI can be expressed in terms of KL divergence as follows:

$$I(S_i; A \mid S_{i-1}) = \mathbb{E}_{S_{i-1}} \left[ D_{KL}(p(S_i \mid S_{i-1}, A) \| p(S_i \mid S_{i-1})) \right]$$

Here, $p(S_i \mid S_{i-1})$ is the marginal distribution of $S_i$ given $S_{i-1}$, which can be approximated by averaging over all actions $a$.

The action selection criterion aims to maximize the KL divergence, effectively choosing the action that maximizes the deviation from the expected next state:

$$a^* = \arg \max_{a \in \{a^1, \ldots, a^z\}} \left( D_{KL}\left( p(s_i \mid s_{i-1}, a) \middle\| \frac{1}{Z} \sum_{k=1}^{Z} p(s_i \mid s_{i-1}, a^z) \right) \right)$$

This criterion is analogous to selecting the action $a$ that maximizes the CMI $I(S_i; A \mid S_{i-1})$. By doing so, it chooses the action that provides the most information about the next state given the current state. This approach effectively enhances the exploration efficiency within the reinforcement learning framework by focusing on actions that exert significant causal influence on the environment.

# B    DATASET

**Synthetic:** Sequences were generated by a self-exciting process with event locations sampled from a mixture of Gaussian distributions which $K = 3$. We set base rate =0.5, event influence=1.2, decay rate=0.5, max time=20, number of sequences =100 and $S = [-1, 1]$, for the Gaussian mixture model,

our parameters are as follows:

Means: $\quad \mu_1 = \begin{pmatrix} 0 \\ 0.1 \end{pmatrix}, \quad \mu_2 = \begin{pmatrix} 0.5 \\ 0.5 \end{pmatrix}, \quad \mu_3 = \begin{pmatrix} -0.5 \\ -0.6 \end{pmatrix}$

Covariances: $\quad \Sigma_1 = \begin{pmatrix} 0.05 & 0.025 \\ 0.025 & 0.05 \end{pmatrix}, \quad \Sigma_2 = \begin{pmatrix} 0.05 & -0.025 \\ -0.025 & 0.055 \end{pmatrix}, \quad \Sigma_3 = \begin{pmatrix} 0.075 & 0.025 \\ 0.025 & 0.05 \end{pmatrix}$

Weights: $\quad w_1 = 0.35, \quad w_2 = 0.35, \quad w_3 = 0.3$

It contains 100 sequences and the average length of sequences is 128.

**Boston Crime:** The Boston dataset, sourced from Kaggle, encompasses sequences of 65 distinct types of crime incidents, including burglary, street crime, and auto theft, gathered over 3 years from the Boston Police Department. Notably, the category "Motor vehicle accident" is what we use in this experiment. Each recorded crime event is marked by a timestamp, location, and other information. After deleting some abnormal values and nan values, we cut it into 168 sequences by time, the average length of sequences is 221.

**911 Call:** The 911Call dataset comprises sequences of emergency phone call records, encompassing EMS, fire, and traffic incidents reported in Montgomery County, PA, spanning from December 2015 to December 2018, with complete location coordinate data. We utilize the category "fire" and cut it into 1100 sequences by time, the average length of these sequences is 91.

**Earthquakes:** Earthquakes in the West US (Latitude from 32.000 to 49.000, Longitude from -125.000 to -112.000 ) from 2000 to 2020 with a magnitude of at least 3.0 are collected from the U.S.Geological Survey. Each record has the magnitude, timestamp, and location coordinate. After deleting some abnormal values, we cut it into 120 sequences by month, the average length of sequences is 89.

## C  ADDITIONAL RESULT

Table 3: Evaluation of performance in predicting time of the next 1 event.

| Model | Synthetic | | Earthquake | | 911 Call | | Crime | |
|---|---|---|---|---|---|---|---|---|
| | Temporal | Spatial | Temporal | Spatial | Temporal | Spatial | Temporal | Spatial |
| Hawkes | 0.327 | - | 1.010 | - | 0.871 | - | 1.432 | - |
| NHP | 0.165 | - | 0.415 | - | 0.227 | - | 0.603 | - |
| THP | 0.161 | - | 0.412 | - | 0.230 | - | 0.601 | - |
| CRLSTPP | 0.152 | 0.089 | 0.393 | 6.608 | 0.197 | 0.182 | 0.581 | 0.127 |

Table 4: Evaluation of performance in predicting time of the next 20 event.

| Model | Synthetic | | Earthquake | | 911 Call | | Crime | |
|---|---|---|---|---|---|---|---|---|
| | Temporal | Spatial | Temporal | Spatial | Temporal | Spatial | Temporal | Spatial |
| Hawkes | 8.216 | - | 19.029 | - | 28.235 | - | 39.245 | - |
| NHP | 4.621 | - | 12.091 | - | 8.926 | - | 16.144 | - |
| THP | 4.347 | - | 11.127 | - | 8.014 | - | 15.924 | - |
| CRLSTPP | 3.293 | 2.017 | 8.024 | 201.283 | 7.021 | 3.713 | 13.038 | 3.282 |

**Hawkes ProcessHawkes (1971):** a self-exciting point process where each event increases the likelihood of future events occurring in the near term.

**The Neural Hawkes Process (NHP)Mei & Eisner (2017):** a LSTM based Hawkes process model.

## D  IMPLEMENTATION DETAILS

We adopt different metrics to evaluate the models. Firstly, we split the sequence to 80% training and 20% test and we evaluate the test set performance. Our task is focusing on the prediction on

short term and long term (which will always cause the compounding prediction error issue ) In this setting, we will use events before time t to predict 1 step or 20 steps after t in the test set. We use MSE to evaluate the results of temporal prediction and spatial prediction, respectively. The details of hyper-parameters used in training are the following. We set K=3,5,5,3 for synthetic, Boston, 911Call, and Earthquake, respectively. The detail hyperparameter selection experiment shown in Table 5. In addition, for most of the datasets, we set sampling number=10, learning rate=2e-4, head number=4, layer number=4, and model dimension D = 512. Our experiment resource including 2 NVIDIA RTX A5000 GPU cards.

Table 5: Hyperparameter selection of Gaussian component number K on synthetic data.

| Model | K=1 | K=3 | K=5 |
|---|---|---|---|
| Temporal - 1 Steps | 0.160 | 0.152 | 0.154 |
| Spatial - 1 Steps | 1.002 | 0.089 | 0.093 |
| Temporal - 20 Steps | 3.472 | 3.293 | 3.409 |
| Spatial - 20 Steps | 3.007 | 2.023 | 2.825 |

