# OpenReview forum: "Causal Reinforcement Learning for Spatio-Temporal Point Processes"
_ICLR.cc/2025/Conference — ICLR 2025 Conference Withdrawn Submission_

### Official Review · Reviewer_HTDv · 2024-10-22

**Soundness:** 2
**Presentation:** 3
**Contribution:** 2
**Rating:** 3
**Confidence:** 4

**Summary:**

The paper proposes a Reinforcement Learning Spatio-Temporal Point Process (CRLSTPP) framework for improved exploration efficiency in generating spatio-temporal event sequences. Extensive experiments across different datasets are conducted to verify its ability.

**Strengths:**

1. The paper is well-written and the idea is clear and simple.
2. Exploration is important in RL.

**Weaknesses:**

[**Overall**] This paper seems to be a pure stack of known methods (STPP, PG, MMD, Mutual Information) without a solid idea that can connect them all. I think the author attempts to use casual reinforcement learning to facilitate exploration in generating STPP, which should be their core contribution. However, the author spends too much time talking about RMDN and MMD, and only has less than 1 page discussing causal action selection (section 4.3) and policy gradient (section 4.5). Furthermore, Sections 4.3 and 4.5 are, in my opinion, completely separate. There is no clue in Section 4.5 as to how the policy given by PG will produce maximal mutual information actions. The author is suggested to clarify how they adapt CMI to the PG loss (include a diagram or pseudocode to illustrate this relationship more clearly) and discuss in depth how CMI will give the casual information to the action in the step. For instance, a comparison between the classic Structural Causal Model (SCM) and introducing CMI to the PG loss.

[**Exploration in RL**] In addition, there are other great RL algorithms like MaxEnt-RL aimed at better exploration. The author also fails to discuss them and make connections to the proposed method (CMI) in the paper. The author is suggested to include at least a discussion section to compare their CMI approach to MaxEnt-RL and other exploration-focused RL algorithms, highlighting key similarities and differences.

[**Structural Causal Model in TPP**] Instead of solely talking about the algorithmic improvement by CMI, the author should emphasize the connection to the data. i.e., TPP.  It is encouraged to discuss how causality is usually modeled in TPP [1,2] and if possible, connect it to the introduced CMI.

[1] Yizhou Zhang et al. Counterfactual Neural Temporal Point Process for Estimating Causal Influence of Misinformation on Social Media

[2] Kimia Noorbakhsh et al. Counterfactual Temporal Point Processes

**Questions:**

1. In Line 269, equation (13), the author attempts to find the action sequences that maximize mutual information. In Line 301, equation (16), the author only uses vanilla PG to update $\pi_{L}$. I am confused about how the action given by the policy gradient in equation (16) will give us the maximum mutual information result in equation (13).

If the author addresses all my concerns above, I am happy to increase my score.

---

### Official Review · Reviewer_wpG6 · 2024-10-24

**Soundness:** 2
**Presentation:** 3
**Contribution:** 2
**Rating:** 5
**Confidence:** 3

**Summary:**

This paper proposes a causal reinforcement learning framework for modeling the spatial temporal point process. They take the time and location of an event as the action of the system and consider that the state consists of the parameters that characterize the underlying spatial temporal distribution. Then the generation of each event can be interpreted as an action taken by stochastic policy. They take the observed sequence to be as the expert policy, and the goal is to use an MMD-based reward to minimize the discrepancy between the learner policy and the expert policy.

**Strengths:**

1. The writing in this paper is well done, with a particularly clear explanation in the methods section.
2. This paper proposes to employ reinforcement learning techniques to address the issue of cascading errors in temporal point processes. These errors, stemming from inaccuracies in a single event, can propagate through subsequent draws, leading to cumulative discrepancies, which are a critical problem to be solved in prediction of temporal point processes.
3. The related work section is comprehensive, showcasing the author's in-depth exploration of background.

**Weaknesses:**

1. The novelty of this paper needs to be enhanced. The initial work of Li. et al. [1] has proposed a method to use reinforcement learning techniques in modeling temporal point processes and treat the generation of each event as the action taken by a stochastic policy. While this paper delves into spatial information and incorporates causal information to enhance action selection, its overall modeling approach and application of RL bear similarities to those in reference [1]. As a result, the novelty and contributions may be somewhat lacking.
2. Some assumptions are not sufficiently robust. For example, the authors assume the next time interval of an event follows Rayleigh distribution, the reason for choosing this distribution for next time interval is explained in detail in [1]. But for the location of the event, why did the author choose a mixture of Gaussian distribution? Another unreasonable assumption is that they assume that the spatial process and the temporal process are independent of each other. However, in certain real-world scenarios, such as criminal activities, the selection of the time and location by suspects often follows distinct patterns. For instance, suspects may opt for nighttime and less secure areas for criminal acts. In such cases, we cannot assume that spatial and temporal processes are independent.
3. Sec 4.3 is not clear. I suggest that the authors explain more about why we need to choose the possible event that results in maximal conditional mutual information deviation from the expectation as Eq.(13). How do you address common clustering phenomena in event sequences?
4. In Sec 5.4, the authors discusses whether the appropriate parametric assumptions are essential for the effectiveness of the point process, which is also my concern. They discuss the phenomenon but lack of experiments to evaluate how the different parametric assumptions affect the performance.
5. Some backgrounds like MMD reward and RMDN are better placed in the preliminaries section.

reference:\
[1] Learning Temporal Point Processes via Reinforcement Learning

**Questions:**

My concern aligns with the points outlined in the weakness section:

1. The novelty and contribution need to be further clarified.
2. Why choose the mixture of Gaussian distribution for the location of the event? Is there any support from any published paper?
3. In Eq.(12), how to determine the decay rate beta? Is it a learnable parameter or not?
4. How to determine the granularity of region partitioning? Since considering the granularity of space region division can impact model performance.
5. In line 261, what is $q$? Does $q$ equal to $n \times n$?
6. Why the incorporation of causal information can help improve action selection? What is the detailed mechanism?
7. Can the author provide some references which support that the spatial process and the temporal process are independent of each other?
8. In the experiment section, I suggest using HYPRO [2] to replace THP. Since HYPRO also considers the cascading error in prediction of temporal point process and this model effectively addresses this issue. Moreover, the proposed model, as an RL model, lacks comparisons with other RL models (such as [1, 3]) in its experiments.
9. The discussion on suitable parametric assumptions is crucial for the efficacy of point processes but lacks experimental validation.

reference:\
[1] Learning Temporal Point Processes via Reinforcement Learning\
[2] HYPRO: A Hybridly Normalized Probabilistic Model for Long-Horizon Prediction of Event Sequences\
[3] Imitation Learning of Neural Spatio-Temporal Point Processes

---

### Official Review · Reviewer_Rq3h · 2024-11-03

**Soundness:** 3
**Presentation:** 3
**Contribution:** 2
**Rating:** 5
**Confidence:** 3

**Summary:**

The paper presents CRLSTPP, a framework combining causal learning with reinforcement learning for modeling spatio-temporal point processes. The reinforcement learning framework mainly follows the setup in Zhu 2021b. The main novelty is the introduction of causal learning to improve exploration efficiency (by prioritize events with higher causal impact).

**Strengths:**

1. The experiments are fairly comprehensive, over multiple real world and synthetic datasets.
2. The approach is well empirically justified, including improvements over baselines and advantage in loss reduction.
3. The visualization is clear and reasonable.
4. The paper is generally well written.

**Weaknesses:**

1. The theoretical foundation seems incomplete. While the paper claims causal learning improves exploration efficiency, the connection between CMI-based action selection and actual causal relationships is not rigorously justified. To be frank, mutual information has nothing to do with causality and the "causal influence" seems more like a statistical correlation.
2. The novelty is limited. Some core methodologies (formula 16, use of MMD rewards, choice of the best action) already present in Zhu 2021b. The use of RMDN and CMI are interesting but also existing techniques, making the paper appears to be an adaptation of Zhu 2021b (with enhanced architectural design to capture long-term temporal dependencies).
3. The math formulation could be improved. Some vectors are bold (z) but others are not (h, w, b, ...), the dimensionality is lacking.

**Questions:**

1. Given your main contribution is the causal reinforcement learning framework for STPP, my main question is that the CMI is not very "causal". To be specific, in the RL settings for STPP, the action is next event space and time, and the state is the spatiotemporal distribution. You imply that choosing the action that causes the maximum state change (next event that causes the largest change in distribution) is the most optimal, and the action is causally linked to previous events. For example, let's think about we have a GPS event sequence for a single agent and we model it using a STPP. If the next event does not change the distribution, can we say it's not causal? Not necessarily. Routine events (like staying at home) may not alter the overall spatiotemporal distribution for the agent too much (especially when there is no location change) but still have stronger true causal relationships. On the other hand, events that alter the distribution (like a car accident) may not be causally linked to previous events. How do you justify the CMI-based action selection truly captures causal relationships rather than just correlations?
2. On the empirical side, the ablation study (comparison with L1 reward) is good but L1 seems overly simple. Are there possible more ablation study on that? Also, the exact formulation for your L1 rewards is missing.
3. The paper claims better exploration efficiency - though I appreciate the empirical evidence, can you provide theoretical guarantees or bounds on the improvement in exploration?

> Figure 2 shows the comparison of reinforcement learning exploration efficiency with/without causal learning.

It's also unclear to me which dataset the loss curve is about. Can you at least show if the faster convergence applies to most datasets in your experiments?

4. Your grid sampling seems defeating the purpose of using STPP (not to worry about the discretization and scale problem; otherwise one can gridfy and use Multivariate Neural Point Process instead). The motivation is still unclear to me. Is it because the pdf is too complicated to sample from? Can't you use more continuous approaches such as rejection sampling? Please provide better explanation.

---

### Note · Authors · 2025-01-23

I have read and agree with the venue's withdrawal policy on behalf of myself and my co-authors.